# The Genome Assembly and Annotation of the Southern Elephant Seal *Mirounga leonina*

**DOI:** 10.3390/genes11020160

**Published:** 2020-02-03

**Authors:** Bo-Mi Kim, Yoon Jin Lee, Jeong-Hoon Kim, Jin-Hyoung Kim, Seunghyun Kang, Euna Jo, Seung Jae Lee, Jun Hyuck Lee, Young Min Chi, Hyun Park

**Affiliations:** 1Unit of Research for Practical Application, Korea Polar Research Institute, Incheon 21990, Korea; bomikim@kopri.re.kr (B.-M.K.); kimjh@kopri.re.kr (J.-H.K.); s.kang@kopri.re.k (S.K.); eunajo@kopri.re.kr (E.J.); junhyucklee@kopri.re.kr (J.H.L.); 2Division of Biotechnology, College of Life Sciences and Biotechnology, Korea University, Seoul 02841; Korea; kukupa08@gmail.com (Y.J.L.); skullcap@korea.ac.kr (S.J.L.); ezeg@korea.ac.kr (Y.M.C.); 3Division of Polar Life Science, Korea Polar Research Institute, Incheon 21990, Korea; jhkim94@kopri.re.kr; 4Polar Sciences, University of Science & Technology, Yuseong-gu, Daejeon 34113, Korea

**Keywords:** southern elephant seal, *Mirounga leonina*, genome assembly: annotation: 10× genomics chromium technology

## Abstract

The southern elephant seal *Mirounga leonina* is the largest phocid seal and one of the two species of elephant seals. They are listed as ‘least concern’ by the International Union for Conservation of Nature (IUCN) Red List of Threatened Species 2015. Here, we have assembled the reference genome for *M. leonina* using the 10× chromium sequencing platform. The final genome assembly of *M. leonina* was 2.42 Gb long, with a contig N50 length of 54 Mb and a maximum length of 111.6 Mb. The *M. leonina* genome contained 20,457 predicted protein-coding genes and possessed 41.51% repeated sequences. The completeness of the *M. leonina* genome was evaluated using benchmarking universal single-copy orthologous genes (BUSCOs): the assembly was highly complete, containing 95.6% of the core set of mammalian genes. The high-quality genomic information on *M. leonina* will be essential for further understanding of adaptive metabolism upon repeated breath-hold dives and the exploration of molecular mechanisms contributing to its unique biochemical and physiological characteristics. The southern elephant seal genome project was deposited at NCBI (National Center for Biotechnology Information) under BioProject number PRJNA587380.

## 1. Introduction

Elephant seals have been highlighted as a crucial animal model for studying their unique behavior, physiology, population dynamics, and geographical distribution. Unlike other ocean-going mammals (e.g., whales, dugongs, manatees), elephant seals have to emerge from the water to rest, molt, mate, and rear pups [1]. Seals undergo a long fasting period, and the energy generated from the blubber-fat allows them stay on land for over a month without food and water. The northern elephant seal is known for losing up to 40% body mass during the prolonged fasting period [2]. Furthermore, seals have evolved to regulate osmotic challenge by producing endogenous water from lipid oxidation and concentration of urine [3].

The elephant seal genus *Mirounga* contains two species, the southern elephant seal *Mirounga leonina* and the northern elephant seal *M. angustirostris*. The southern elephant seal is the largest species in the clade Pinnipedia, as adult males can weigh up to four tons and grow up to 5.8 m in length. Four distinct populations were investigated in the circumpolar regions and sub-Antarctic islands [4,5]. Southern elephant seals have evolved to be able to deep- and long- dive for foraging. During predation, they spend most of their time underwater to prey on squid, mollusks, krill, and cephalopods, and they can undertake lengthy dives from 30~45 min to over 2 h and dive more than 2000 m in depth [6]. The enlarged eyes and sensitive vibrissae whiskers play a crucial role in predation in the deep sea [6].

*M. leonina* was almost exterminated by indiscriminate slaughter to harvest oil from the blubber in the early 19th century. Thus, they were listed as ‘vulnerable’ under the predecessor to the Environment Protection and Biodiversity Conservation Act 1999 (EPBC Act). Hunting was regulated by the International Convention for the Conservation of Antarctic seals and the Convention on Antarctic Marine Living Resources to conserve *M. leonina* individuals and their population, and population size has been recovered. They are currently listed as ‘least concern’ by the International Union for Conservation of Nature (IUCN) Red List of Threatened Species 2015 (e.T13583A45227247, http://www.iucnredlist.org, accessed on: 12. Dec. 2014). The southern elephant seal has encountered several threats, such as the depletion of prey stocks owing to intensive fishing in Antarctica, drastic fluctuations in habitat by climate change, and low genetic diversity [7]. Therefore, the consistent monitoring and highly regulated conservation of *M. leonina* populations are strongly required.

Genome information of *M. leonina* could help to understand the unique physiological and metabolic adaptations, such as the ability to spend large amounts of time in the sea as a mammal, survive in a frigid environment, perform repeated breath-hold dives, engage in deep-diving and long-ranging predation, tolerate routine hypoxia, and fast during the nursing period. In this study, the genome of the southern elephant seal was sequenced and analyzed for the first time in elephant seals. The genome of *M. leonina* was effectively sequenced from a small blood sample using the 10× chromium sequencing platform to diminish potential stress during sampling. This study will contribute to preserving the species by understanding its physiology and molecular mechanisms.

## 2. Materials and Methods 

The genomic DNA of the southern elephant seal, *Mirounga leonina*, (Figure 1) was extracted from the blood specimen as described in our previous study [8]; approximately 1 mL of blood sample was collected from a single elephant seal on King George Island, South Shetland Islands, Antarctica. The blood sample (200 µL) was used to obtain high molecular weight gDNA using the QIAGEN MagAttract HMW DNA kit (QIAGEN, Germantown, MD, USA) according to the manufacturer’s protocol. The quality and quantity of the gDNA were analyzed using a 5400 Fragment analyzer (Agilent Technologies, Santa Clara, CA, USA) and Qubit 2.0 Fluorometer (Invitrogen, Life Technologies, Carlsbad, CA, USA). DNA libraries for the southern elephant seal individuals were generated using the 10× Genomics Chromium technology according to the manufacturer’s instructions. Gel bead-in-emulsions (GEMs) were created from a library of Genome Gel Beads combined with 1.5 ng of gDNA in a Master Mix and partitioning oil, using the 10× Genomics Chromium Controller instrument with a micro-fluidic Genome chip (PN-120257). The GEMs were then subjected to an isothermal incubation step. Bar-coded DNA fragments were extracted and underwent Illumina library construction, as detailed in the Chromium Genome Reagent Kits Version 2 User Guide (PN-120258). Library yield was measured through the Qubit dsDNA HS assay kit (Thermo Fisher Scientific, Waltham, MA, USA). Library fragment size and distribution were measured using an Agilent 2100 Bioanalyzer High Sensitivity DNA chip (Agilent, Santa Clara, CA, USA). The DNA was sequenced on a NovaSeq with a 2 × 250 bp read metric, and ∼1.6 billion paired-end (PE) reads were generated (Table 1).

The de novo genome assembly was performed using the paired-end sequence reads from the partitioned library as input for the Supernova assembler (RRID:SCR 016756) v2.1.1 (10× Genomics, San Francisco, CA, USA) [9] with default parameters.

We built a de novo repeat library using RepeatModeler v1.0.3, (RepeatModeler, RRID: SCR 015027) [10], including the RECON (RECON, RRID:SCR 006345) [10] and RepeatScout v1.0.5 (RepeatScout, RRID:SCR 014653) [11] software with default parameters. Tandem Repeats Finder [12] was used to predict consensus sequences, classification information for each repeat, and tandem repeats, including simple repeats, satellites, and low-complexity repeats.

We annotated the genome with MAKER (RRID:SCR_005309) pipeline [13], which implements both ab initio prediction and homology-based gene annotation. Ab initio gene prediction was performed using SNAP (SNAP, RRID:SCR 002127) [14] and Augustus (Augustus: Gene Prediction, RRID:SCR 008417). MAKER was initially run in est2genome mode, which was based on transcripts for *M. leonina* generated from the previous results [8]. As further evidence for the annotation, reference proteins from other sequenced mammal species (*Arctocephalus gazelle*, *Leptonychotes weddellii, Odobenus rosmarus, Tursiops truncates, Orcinus orca, Trichechus manatus latirostris, Canis lupus familiaris, Felis catus, Equus caballus*, and *Homo sapiens*) were used in the analysis. The Exonerate software package was used to polish the MAKER alignments, which provides integrated information for the SNAP software program (SNAP, RRID:SCR 007936). The Infernal software package ver. 1.1 (INFERNAL, RRID:SCR 011809) [15] and covariance models (CMs) from the Rfam database v12.1 (Rfam, RRID:SCR 007891) [16] were used to identify other non-coding RNAs. The putative tRNA genes were identified using tRNAscan-SE v1.3.1 (tRNAscan-SE, RRID:SCR 010835) [17], which uses a CM that scores candidates based on their sequence and predicted secondary structures. Functional annotations were conducted by aligning them to the NCBI (National Center for Biotechnology Information) non-redundant protein (nr), Swissprot (Swissprot, RRID:SCR_002380) [18], TrEMBL (TrEMBL, RRID:SCR_002380) [18] using BLAST v2.2.31 [19] with a maximal e-value of 1e-5, and to the Pfam database (Pfam, RRID:SCR_004726) [20] using HMMer V3.0 [21].

This study including sample collection and experimental research conducted on these animals was performed according to the law on activities and environmental protection to Antarctic approved by the Minister of Foreign Affairs and Trade of the Republic of Korea.

## 3. Results and Discussion

The final assembled southern elephant seal, *Mirounga leonina*, genome obtained a 2.42 Gb genome (GC content: 41.51%) estimated the heterozygosity to be 0.301% with an N50 scaffold length of 54.233 Mb and a maximum scaffold length of 111.625 Mb (Table 2 and Appendix A). The number of scaffolds in the southern elephant seal genome assembly was 1.115, and 54 scaffolds were over 10 Mb long and occupied 90.6% of our assembly (Appendix A.). Genome completeness was estimated with Benchmarking Universal Single-copy Orthologs (BUSCO) v3.0. (RRID:SCR 015008, version 3.0) by using the vertebrata_odb9 database [22]. Of the 2586 total BUSCO groups searched, 2472 BUSCO core genes were completely, and 49 were partially identified, leading to a total of 98.1% of BUSCO genes being found in the *M. leonina* genome (Appendix A).

The analysis revealed that 41.51% of the assembled *M. leonina* genome consisted of repeat sequences, of which 33.28% were transposable elements (TE), including LINEs (22.12%), SINEs (6.06%), LTRs (3.06%), and DNA transposons (1.81%) (Table 3). The Kimura divergence (κ-values) estimates the age and transposition history of TEs [23]. The LINEs are the most abundant transposable elements. Kimura distances (κ-values) were calculated for all TE copies of each element in order to estimate the “relative age” and transposition history of TEs [24]. The calculated Kimura divergence for all the TE copies of *M. leonina* is shown (Figure 2) and strongly shaped by LINEs (κ-values ≤5), which means that transposable elements, and LINEs in particular, are a recent development in the southern elephant seal genome due to very similar copies (low κ-values) are indicative of rather recent activity.

A total of 20,457 protein-coding genes in the M. leonina genome were annotated based on the combination of ab initio gene prediction, homology search, and transcript mapping. The total length of exons occupied 30.5 Mb, with an average of 8.2 exons per gene (Table 4).

The functional classification of Gene Ontology (GO) categories (Gene Ontology, RRID:SCR_002811) was performed using the BLAST2GO v5.25 pipeline (Blast2GO, RRID:SCR_005828) [25]. A Kyoto Encyclopedia of Genes and Genomes (KEGG, RRID:SCR_001120) [26] pathway annotation analysis was performed using the KEGG Automatic Annotation Server (KAAS), and EuKaryotic Orthologous Groups [27] were annotated with the KOG database proteins using BLASTp v2.2.31 with a maximal e-value of 1e-5. As a result, totally, 19,439 genes were annotated by a minimum of one database (Table 4). A total of 3354, 15,354, and 11,501 genes were annotated by the gene ontology (GO), Kyoto Encyclopedia of Genes and Genomes (KEGG) databases, and Eukaryotic Orthogous Groups (KOG), respectively (Appendix A). The genome browser, local blast database and assembly data are accessible on https://antagen.kopri.re.kr/project/project.php, accessed on: 01.01.2020.

Synteny analyses of chromosomes between California sea lion (*Zalophus californianus*) reference genome and southern elephant seal were performed using SyMAP v3.4 [28]. Southern elephant seal assembly is highly contiguous compared to the 18 chromosomes of the California sea lion (Figure 3 and Appendix A). A total of 2.22 Gb of southern elephant seal assembled genome were mapped to California sea lion chromosomes. OrthoVenn2 [29] was used to identify paralogy and orthology in annotated proteins of the southern elephant seal and other marine mammals. The deduced southern elephant seal proteins have 13,473 orthologous groups (containing 14,335 genes of the southern elephant seal) within five Pinnipedia species, and 148 orthogroups (containing 353 genes) are only for southern elephant seal (Figure 4).

The southern elephant seal assembly shows that a high-quality reference genome from 10× linked reads sequence will be essential for the further understanding of adaptive metabolism upon repeated breath-hold dives and the exploration of molecular mechanisms contributing to its unique biochemical and physiological characteristics. Additionally, our genomic data will provide valuable genetic resources for evolutionary studies regarding the divergence of pinnipeds.

## Figures and Tables

**Figure 1 genes-11-00160-f001:**
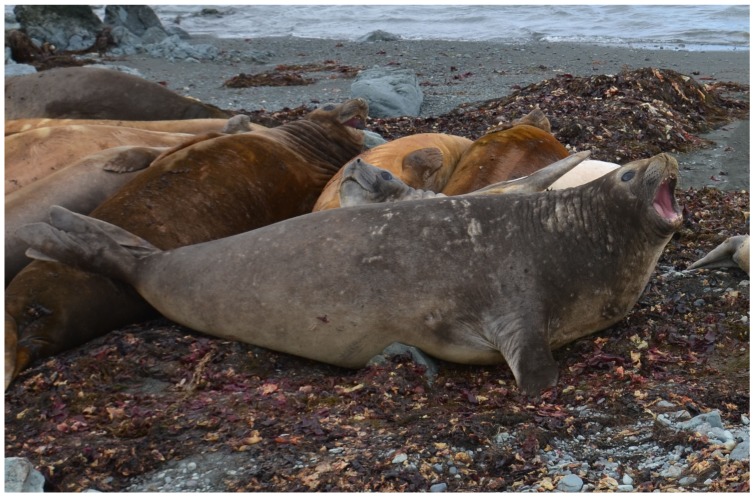
Example of *Mirounga leonina*, southern elephant seal. Photo by Hyun Park.

**Figure 2 genes-11-00160-f002:**
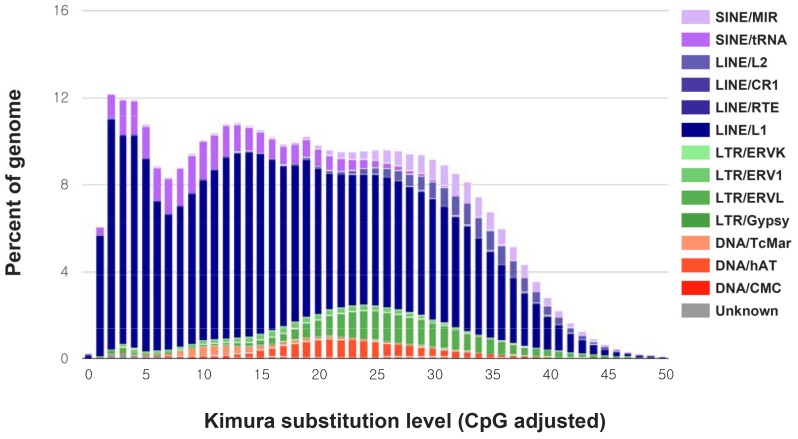
Kimura distance-based copy divergence analysis of transposable elements in the *M. leonina* genome. Graphs represent genome coverage (Y-axis) for each type of TE (DNA transposons, SINE, LINE, and LTR retrotransposons).

**Figure 3 genes-11-00160-f003:**
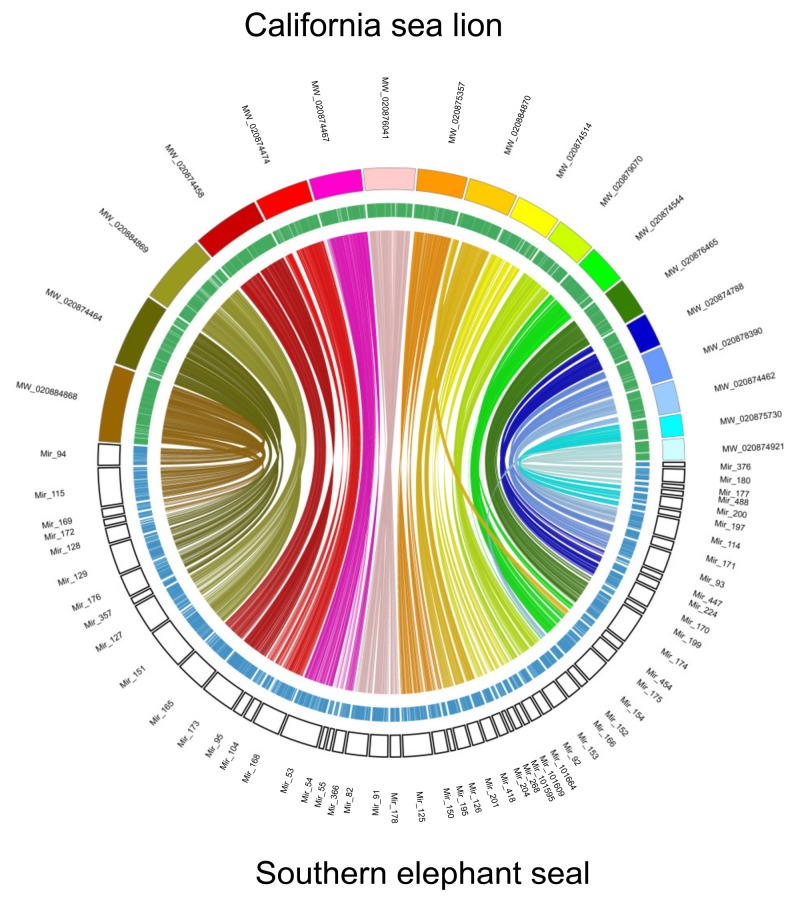
Synteny blocks between Southern elephant seal scaffolds and 19 chromosomes of California sea lion (assembly zalCal2.2). Southern elephant seal scaffolds over 10 Mb in length were selected. Alignment was accomplished with SyMAP v3.4. The color blocks represent California sea lion chromosomes and empty blocks represent Southern elephant seal scaffolds. Green and blue bars represent genes of California sea lion and Southern elephant seal, respectively. Connections within the circle represent alignment between the two assemblies.

**Figure 4 genes-11-00160-f004:**
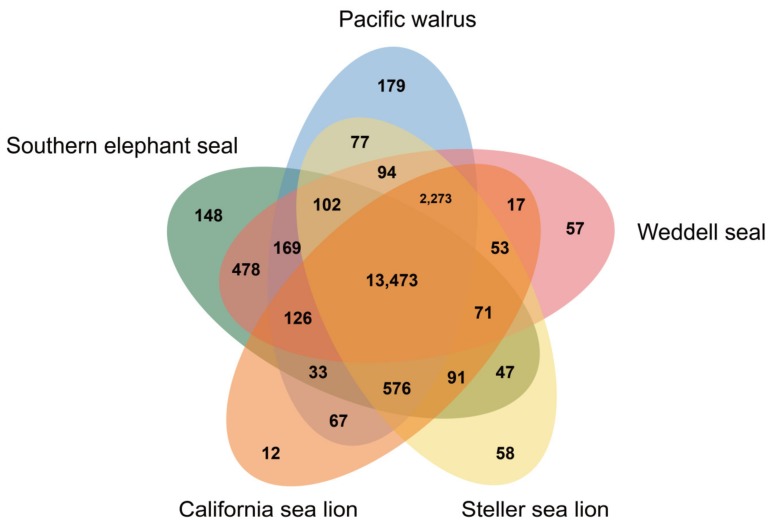
Venn diagram representing paralogous and orthologous groups between Southern elephant seal (*Mirounga leonina*), Pacific walrus (*Odobenus rosmarus*), Weddell seal (*Leptonychotes weddellii*), Steller sea lion (*Eumetopias jubatus*) and California sea lion (*Zalophus californianus*) obtained with OrthoVenn2.

**Table 1 genes-11-00160-t001:** Sequencing data generated for *Mirounga leonina* genome assembly and annotation.

	Number of Reads	Total Read Bases
Genome	1,606,582,076	242,593,893,476
Transcriptome	80,958,400	19,430,016,000

**Table 2 genes-11-00160-t002:** Statistics for *Mirounga leonina* genome assembly.

Assembly	Supernova 2.0
Number of scaffolds	1115
Total size of scaffolds	2,417,339,903
Longest scaffolds	111,625,095
N50 scaffolds length	54,232,831
Number of scaffolds >10M	54
Gap (%)	0.65

**Table 3 genes-11-00160-t003:** Statistics for annotated *Mirounga leonina* transposable elements.

	Number of Elements	Length (bp)	Percentage of Sequence (%)
**SINEs:**	**888,301**	**148,028,385**	**6.06**
MIRs	254,250	35,222,085	1.44
**LINEs:**	**1,624,363**	**540,626,278**	**22.12**
LINE1	1,473,069	508,962,965	20.83
LINE2	150,067	31,135,376	1.27
L3/CR1	634	81,165	0.00
**LTR elements:**	**268,713**	**74,825,384**	**3.06**
ERVL	81,307	29,432,080	1.20
ERVL-MaLRs	134,246	29,421,479	1.20
ERV_classI	52,451	15,768,764	0.65
ERV_classII	370	158,839	0.01
**DNA elements:**	**261,769**	**44,140,597**	**1.81**
hAT-Charlie	146,299	24,083,632	0.99
TcMar-Tigger	43,576	10,220,361	0.42
**Unclassified:**	**20,630**	**5,747,808**	**0.24**
**Total interspersed repeats:**		**813,368,452**	**33.28**
**Small RNA:**	**637,161**	**113,111,897**	**4.63**
**Satellites:**	**1867**	**85,913**	**0.00**
**Simple repeats:**	**618,437**	**26,612,557**	**1.09**
**Low complexity:**	**99,311**	**5,218,008**	**0.21**

Bold are total of each classes of transposable elements.

**Table 4 genes-11-00160-t004:** *Mirounga leonina* genome annotation statistics.

Annotation Database	Annotated Number	Percentage (%)
No. of genes	20,457	
nr annotation	19,901	97.3
GO annotation	3109	15.2
KEGG annotation	11,501	56.2
KOG annotation	15,498	75.8
Pfam annotation	16,733	81.8
Swissprot annotation	19,653	96.1
TrEMBL annotation	17,600	86.0
	**Count**	**Length Sum (bp)**
Exon	168,375	30,497,777
CDS	167,919	28,718,340

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
