# Peer review of "The Genome Assembly and Annotation of the Southern Elephant Seal Mirounga leonina"

_genes, 2020, doi:10.3390/genes11020160_

Round 1

Reviewer 2 Report

This article describes the assembly, gene prediction and annotation of the genome of the southern elephant seal Mirounga leonina. This work has, in principle, been well carried out and the genome will be a robust dataset for a range of further investigations.

There is only one major problem with this article, and it is that none of this data (other than the raw reads upon which the assembly is based at SRR10445712) has been made available. PRJNA587380 contains only links to the SRA reads and the mitochondrial genome.

Without the public release of the genome assembly, the protein and gene sequence data, and a file describing the location of these genes on the genome (gff/gtf or similar) this work cannot be reviewed correctly, and the paper cannot be published in its present form. It would also be useful to release full data regarding the location and sequence of repeat elements, and perhaps hard/soft masked versions of the genome.

If these data are made available in full in a permanent repository, there is little to preclude publication. Without it, the paper is only an advertisement. I have also suggested some minor revisions below.

Is ethical approval necessary for dealing with vertebrate samples in Korean universities? If so, show ethical approval board approval number, and if not, this should be noted (see lines 121-123, animal rights are not noted). 

Minor comments:

Was heterozygosity taken into account at any point in the assembly process? Was it flattened after assembly? Can any statistics be provided on heterozygosity levels in the genome?

Line 64: comma before "such as"

Line 69: "This study will..." (not would)

Fig 1: where is this image from? Perhaps give photographer name.

Line 99: built (not build)

Line 109 on: Scientific names need to be italicised.

Line 117 on: provide the results of functional annotation somewhere (supplementary file?)

Line 125: obtained (not "was obtained")

Line 154: Italics for M. leonina.

Fig 3 is very fuzzy and difficult to read

Fig 4: Southern is spelled incorrectly

Line 178: "which confirmed" not "which confirming"

Line 178-179: mapping shows that these data are homologous, not that assembly is good. Revise

Fig 5 legend: Co-linear is mis-spelled

It would be good to compare assembly metrics with chromosomal number. Is a karyotype available for Mirounga leonina? If so, commenting in lines 174-177 about relative scaffold length and comparison to California Sea Lion chromosome number would add much context to this article.

Reference 25: Capitals for Kyoto

Round 2

Reviewer 2 Report

The authors have adequately addressed my concerns.

Suggested tweaks:

The genomescope figure provided to me should be placed in the supplementary data. Consider noting the heterozygosity in-text

Consider also placing the assembly on an independent repository, for data security and ease of access

Line 29: are accessible (not were accessible)

California is mis-spelled in Fig 3

I think the authors mis-understood my reference to the karyotype. This is available (2n = 34) and is interesting to compare with the California sea lion. The largest sea lion scaffold corresponds to two of the scaffolds in this work, for example. https://www.ncbi.nlm.nih.gov/pubmed/436448
